# Evaluation of the United States COVID-19 vaccine allocation strategy

Md Rafiul Islam[1], Tamer Oraby[2], Audrey McCombs[3], Mohammad Mihrab Chowdhury[4], Mohammad Al-Mamun[5], Michael G. Tyshenko[6], Claus Kadelka[1]*

1 Department of Mathematics, Iowa State University, Ames, IA, United States of America, 2 School of Mathematical and Statistical Sciences, The University of Texas Rio Grande Valley, Edinburg, TX, United States of America, 3 Department of Statistics, Iowa State University, Ames, IA, United States of America, 4 Department of Mathematics and Statistics, Texas Tech University, Lubbock, TX, United States of America, 5 Department of Pharmaceutical Systems and Policy, West Virginia University, Morgantown, WV, United States of America, 6 McLaughlin Centre for Population Health Risk Assessment, Faculty of Medicine, University of Ottawa, Ottawa, ON, Canada

* ckadelka@iastate.edu

## Abstract

### Background

Anticipating an initial shortage of vaccines for COVID-19, the Centers for Disease Control (CDC) in the United States developed priority vaccine allocations for specific demographic groups in the population. This study evaluates the performance of the CDC vaccine allocation strategy with respect to multiple potentially competing vaccination goals (minimizing mortality, cases, infections, and years of life lost (YLL)), under the same framework as the CDC allocation: four priority vaccination groups and population demographics stratified by age, comorbidities, occupation and living condition (congested or non-congested).

### Methods and findings

We developed a compartmental disease model that incorporates key elements of the current pandemic including age-varying susceptibility to infection, age-varying clinical fraction, an active case-count dependent social distancing level, and time-varying infectivity (accounting for the emergence of more infectious virus strains). The CDC allocation strategy is compared to all other possibly optimal allocations that stagger vaccine roll-out in up to four phases (17.5 million strategies). The CDC allocation strategy performed well in all vaccination goals but never optimally. Under the developed model, the CDC allocation deviated from the optimal allocations by small amounts, with 0.19% more deaths, 4.0% more cases, 4.07% more infections, and 0.97% higher YLL, than the respective optimal strategies. The CDC decision to not prioritize the vaccination of individuals under the age of 16 was optimal, as was the prioritization of health-care workers and other essential workers over non-essential workers. Finally, a higher prioritization of individuals with comorbidities in all age groups improved outcomes compared to the CDC allocation.

**Data Availability Statement:** Data and all relevant code is available at GitHub: https://github.com/ckadelka/COVID19-CDC-allocation-evaluation.

**Funding:** The author(s) received no specific funding for this work.

**Competing interests:** The authors have declared that no competing interests exist.

## Conclusion

The developed approach can be used to inform the design of future vaccine allocation strategies in the United States, or adapted for use by other countries seeking to optimize the effectiveness of their vaccine allocation strategies.

## Introduction

Prior to the U.S. Food and Drug Administration's Emergency Use Authorization of COVID-19 vaccines, the Centers for Disease Control and Prevention (CDC), guided by the federal Advisory Committee on Immunization Practices (ACIP), ranked population groups by priority for initial vaccination roll-out, based on available scientific evidence, the feasibility of different implementation strategies, and ethical considerations [1, 2]. Phase 1a included health care personnel and long-term care facility (LTCF) residents; Phase 1b included frontline essential workers and individuals 75 years old or older; and Phase 1c included other essential workers, 16–64 year olds with high-risk conditions, and 65–74 year olds. Phase 2 included 16–64 year olds without high-risk conditions or comorbidities [2].

COVID-19 vaccine prioritization strategies have been studied in many ways, by using deterministic differential equation models [3–8], agent-based models [9, 10], network models [11, 12], and various other approaches [13, 14], as well as by considering ethical factors [15, 16]. Most studies focused solely on age and age-dependent disease behavior [6–8, 14]. Some studies included further characteristics such as occupation (e.g., distinguishing essential workers) [3, 4, 13], comorbidities [5], and contact patterns [9, 10]. However, none of the model-based studies considered together all characteristics included in the CDC recommendations (age, occupation, comorbidity status, and living condition). More importantly, none of the studies investigated all possible vaccine allocation strategies; rather, a small number of strategies were typically selected for comparison based on expert opinion. The goal of this study was to directly evaluate the CDC recommendation by comparing it to all potentially optimal allocation strategies that stagger the vaccine roll-out in up to four phases (17.5 million strategies).

## Methods overview

### Model design

To achieve an accurate evaluation of the CDC vaccine allocation strategy, we developed a compartmental disease model that stratifies the U.S. population by all characteristics included in the CDC recommendations. Using recent U.S. census data, we divided the population into different sub-populations based on age (four classes: $0 - 15$, $16 - 64$, $65 - 74$, 75+ years old), comorbidity status (two classes: none or at least one known risk factor associated with more severe COVID-19 disease other than age), job type of the working-age population (four classes: healthcare workers, frontline essential workers, other essential workers, and all remaining people), and living situation of individuals 65 and older (two classes: congested living or not). The model also takes into account various important components of the COVID-19 pandemic (Fig 1): age-dependent susceptibility to infection and severity of disease [17, 18] (S1 Fig); age- and comorbidity status-dependent case fatality rates [19]; average rates of contact that differ with age, profession, and living conditions [20–22] (S2 Fig); population-wide social distancing levels that depend on the active number of cases (S3 Fig); the exact speed of the U.S. vaccine roll-

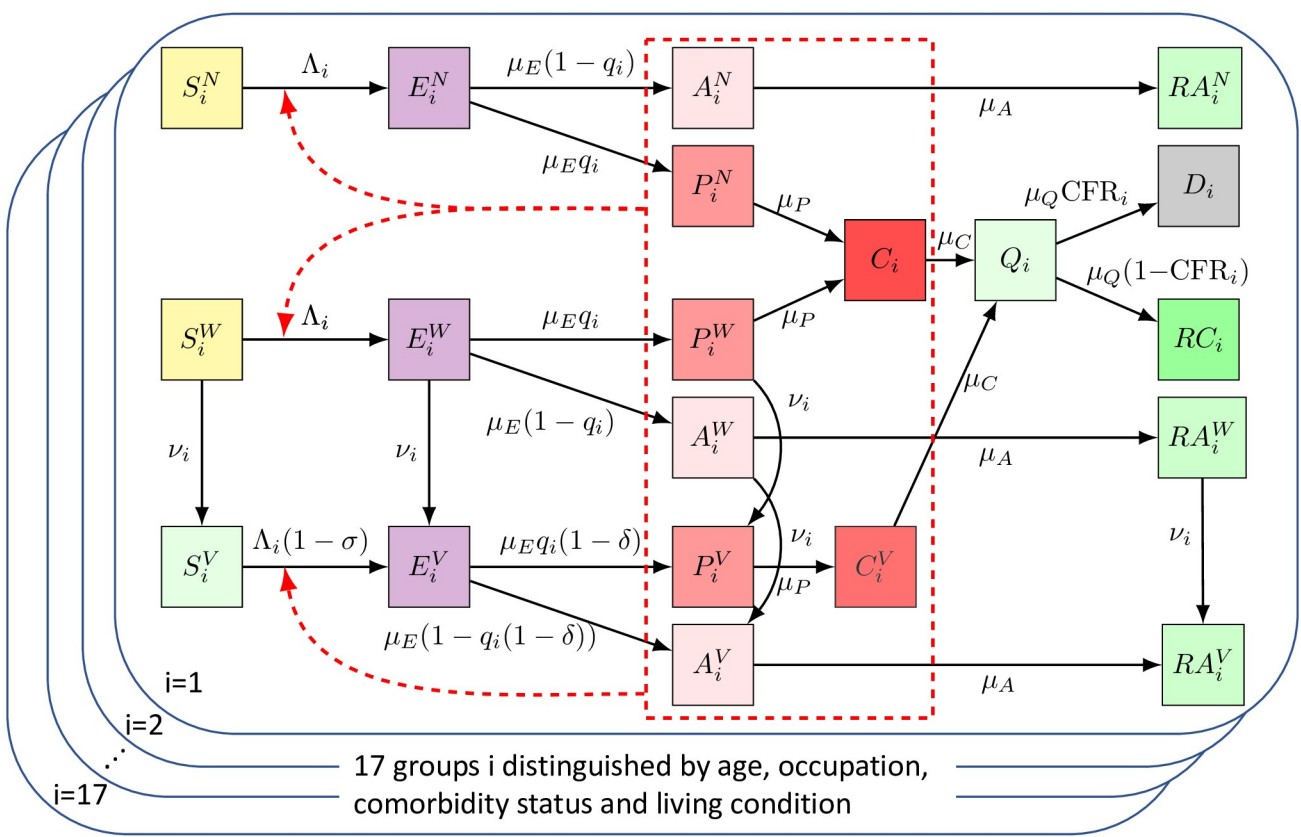

**Fig 1. Overview of the model.** Upon infection, susceptible individuals (left most column) transition through the various disease compartments (middle columns) until reaching a final compartment (death or recovery; right most columns). All pre-clinical, clinical and asymptomatic individuals may cause new infections (red dashed box). A detailed description of the various compartments and parameters can be found in Table 1 and the Detailed Methods section.

out and vaccine hesitancy (S4 Fig); and the emergence of more transmissible virus variants over time [23] (S5 Fig).

We derived most model parameters from the literature (Table 1), and employed an elitist genetic algorithm to estimate the remaining parameters by fitting the model to observed cumulative deaths and cases between December 14, 2020 and April 29, 2021 [24] (S1 Table and S6 Fig). The model was implemented in Python 3.8 using the open source JIT compiler numba for improved run time.

## Study design and outcomes

There are $4^{17} \approx 1.7 \times 10^{10}$ ways to allocate 17 sub-populations into four phases, the number of phases specified by the CDC recommendations. A large number of these allocations can be ruled out as non-optimal, for the following reasons. First, any strategy that recommends vaccination of a specific group of people (i.e., certain age, occupation and living condition) without comorbidities before vaccination of that same group with comorbidities can be improved by switching the two phase assignments, as this leads to a reduction of deaths while not changing case numbers. This rule reduces the number of feasible allocations to $4 \times 10^8$. Second, individuals of the same age and the same comorbidity status but with jobs with higher contact rates or in congested living conditions should never be vaccinated later than people with a lower-contact job or not in congested living conditions, as this leads to a reduction of cases and a

**Table 1. Model parameters, description and sources.**

| Parameter | Description | Value | Source |
|---|---|---|---|
| $N_i$ | number of people in sub-population $i$ | see Table 2 | [25] |
| $X_{ij}$ | average daily number of contacts a person in sub-population $i$ has with sub-population $j$ | see S2G Fig | [21, 22] |
| $c$ | log10 value of active cases at which overall contacts are reduced by 50% | $c = 4.0346$ (see S1 Table for fitted values used in the sensitivity analysis) | fitted (see Model calibration) |
| $k$ | sensitivity of contact reduction to changes in active cases (shape of the Hill function) | $k = 5.0266$ (see S1 Table for fitted values used in the sensitivity analysis) | fitted (see Model calibration) |
| $\beta_i$ | age-dependent susceptibility to infection | see S1 Table | fitted (see Model calibration) |
| $1/\mu_E$ | incubation period | 3.7 days | [26] |
| $q_i$ | age-dependent clinical fraction | varied, see S1 Fig | [27] |
| $1/\mu_A$ | average time of virus spread by truly asymptomatic individuals | 5 days | [17] |
| $1/\mu_P$ | average time of virus spread before symptom onset | 2.1 days | [17] |
| $1/\mu_C$ | average time of virus spread after symptom onset | 2.723 days | estimated from CDC raw data |
| $1/\mu_Q + 1/\mu_C$ | average time between symptom onset and possible death | 22 days | estimated from U.S. deaths and case counts [28] |
| $CFR_i$ | sub-population-dependent case fatality ratio | see Case fatality rates | calculated from [29, 30] |
| $f_A$ | relative contagiousness of truly asymptomatic individuals | 75% (25% and 100% in sensitivity analysis) | [27] |
| $f_V$ | relative contagiousness of vaccinated individuals | 50% (0% and 100% in sensitivity analysis) | no data |
| none | vaccine hesitancy | 30% | [31, 32] |
| $\xi(t)$ | daily number of available vaccines | see S4 Fig | [33] |
| none | vaccine effectiveness: reduction of symptomatic infections among vaccinated (compared to non-vaccinated) | 90% | [34] |
| $\sigma$ and $\delta$ | reduction in infections and symptomatic infections (when infected) among vaccinated (compared to non-vaccinated) individuals | 70% and 66.7% (varied such that $1 - (1 - \sigma)(1 - \delta) = 90\%$ in sensitivity analysis) | [35] |

subsequent reduction in deaths. This rule reduces the number of feasible, potentially optimal allocations to $1.75 \times 10^7$.

In a novel global optimization approach, we compared all these 17.5 million potentially optimal vaccine allocation strategies that stagger the vaccine roll-out in up to four phases using four primary outcome metrics: total deaths, total cases, total infections (symptomatic and asymptomatic), and total years of life lost (YLL) at the end of 2021, slightly more than one year after the beginning of the public vaccine roll-out in the United States (Table 2). We computed all strategies that are Pareto-optimal and compared with the CDC allocation strategy (Fig 2). Pareto-optimal strategies cannot be improved in one metric without obtaining a worse outcome in another metric. As a secondary outcome measure, we investigated how deaths were distributed across different age groups as one element of health equity, as equitable vaccine allocation has received attention both from government agencies and the media [36].

## Results

### Evaluation of vaccine allocation strategies

Overall, the CDC strategy performed well in all metrics but never optimally (Table 2). According to the established model, there were other allocations that resulted in 0.19% lower mortality, 0.97% lower YLL, 4.0% fewer cases and 4.09% fewer infections. Prioritizing the vaccination of the working age population generally led to fewer cases and infections at the expense of higher deaths and YLL, highlighting the anticipated trade-off in multi-objective decision making. Indeed, pairwise Spearman correlations between the four metrics (Fig 2A) revealed that it is not possible to find a single allocation strategy that is optimal under each objective.

**Table 2. Comparison of CDC and optimal vaccine allocation strategies.** For each sub-population (characteristics and population sizes defined in the left columns) and each objective (top row), the priority phase corresponding to the optimal allocation strategy is shown. At the bottom, absolute and relative outcomes are compared for the CDC allocation and all optimal allocation strategies.

| Age | Job / living situation | Comorbidity | Number of people [millions] | Sub-population ID in model | CDC allocation | fewest deaths [thousands] | lowest YLL [millions] | fewest cases [millions] | fewest infections [millions] |
|---|---|---|---|---|---|---|---|---|---|
| 0-15 | NA | NA | 64.71 | 1 | 4 | 4 | 4 | 4 | 4 |
| 16-64 | healthcare workers | no | 13.29 | 2 | 1 | 1 | 1 | 1 | 1 |
| | healthcare workers | yes | 7.71 | 3 | 1 | 1 | 1 | 1 | 1 |
| | frontline essential workers | no | 18.98 | 4 | 2 | 2 | 2 | 2 | 2 |
| | frontline essential workers | yes | 11.02 | 5 | 2 | 2 | 2 | 2 | 2 |
| | other essential workers | no | 12.66 | 6 | 3 | 3 | 3 | 2 | 2 |
| | other essential workers | yes | 7.34 | 7 | 3 | 3 | 2 | 2 | 2 |
| | remaining people | no | 87.61 | 8 | 4 | 4 | 4 | 3 | 3 |
| | remaining people | yes | 50.85 | 9 | 3 | 3 | 3 | 3 | 3 |
| 65-74 | congested living | no | 0.28 | 10 | 1 | 2 | 3 | 3 | 3 |
| | congested living | yes | 0.76 | 11 | 1 | 1 | 2 | 3 | 3 |
| | remaining people | no | 8.20 | 12 | 3 | 3 | 4 | 4 | 4 |
| | remaining people | yes | 22.34 | 13 | 3 | 3 | 3 | 4 | 4 |
| 75+ | congested living | no | 0.39 | 14 | 1 | 3 | 3 | 3 | 4 |
| | congested living | yes | 1.57 | 15 | 1 | 1 | 2 | 3 | 4 |
| | remaining people | no | 4.07 | 16 | 2 | 3 | 4 | 4 | 4 |
| | remaining people | yes | 16.47 | 17 | 2 | 2 | 3 | 4 | 4 |
| **Respective outcome of specific allocation** | | | | CDC | 652 | 11.6 | 38.1 | 56.4 |
| | | | | fewest deaths | 651 | 11.6 | 37.9 | 56.2 |
| | | | | lowest YLL | 657 | 11.5 | 37.3 | 55.3 |
| | | | | fewest cases | 688 | 11.8 | 36.6 | 54.2 |
| | | | | fewest infections | 695 | 11.9 | 36.6 | 54.2 |
| **% difference in outcome between specific and respective optimal allocation** | | | | CDC | 0.19 | 0.97 | 4 | 4.07 |
| | | | | fewest deaths | 0 | 0.67 | 3.64 | 3.74 |
| | | | | lowest YLL | 0.88 | 0 | 1.9 | 2.08 |
| | | | | fewest cases | 5.75 | 2.44 | 0 | 0.01 |
| | | | | fewest infections | 6.73 | 2.84 | 0.03 | 0 |

Rather, there exist several Pareto-optimal solutions to the multi-objective optimization problem. The Pareto frontier consists of all strategies that cannot be further improved in one objective without obtaining a worse outcome in another objective. The two-dimensional Pareto frontier based on deaths and cases reveals that the strategy chosen by the CDC was almost optimal (Fig 2B), and could only be outperformed by a few strategies (S2 Table). All these Pareto-dominant strategies put more emphasis on a differential phase assignment of individuals with and without comorbidities; that is, vaccinating individuals with COVID-19 risk factors earlier.

Equitable allocation in the vaccine roll-out has received attention both from government agencies and the media [36]. Here, we investigated how deaths are distributed across different age groups as one element of health equity. Interestingly, the allocation strategy that minimizes overall mortality also leads to a more even distribution of deaths across the age groups, compared to other single-objective optimal allocations or the CDC allocation (Fig 2C). Across all 17.5 million investigated allocations, the most age-equitable allocation strategy, measured using the entropy of the mortality distribution across the four age groups, performed poorly in all other objectives, as did an unstructured vaccine roll-out without phases (S7 Fig).

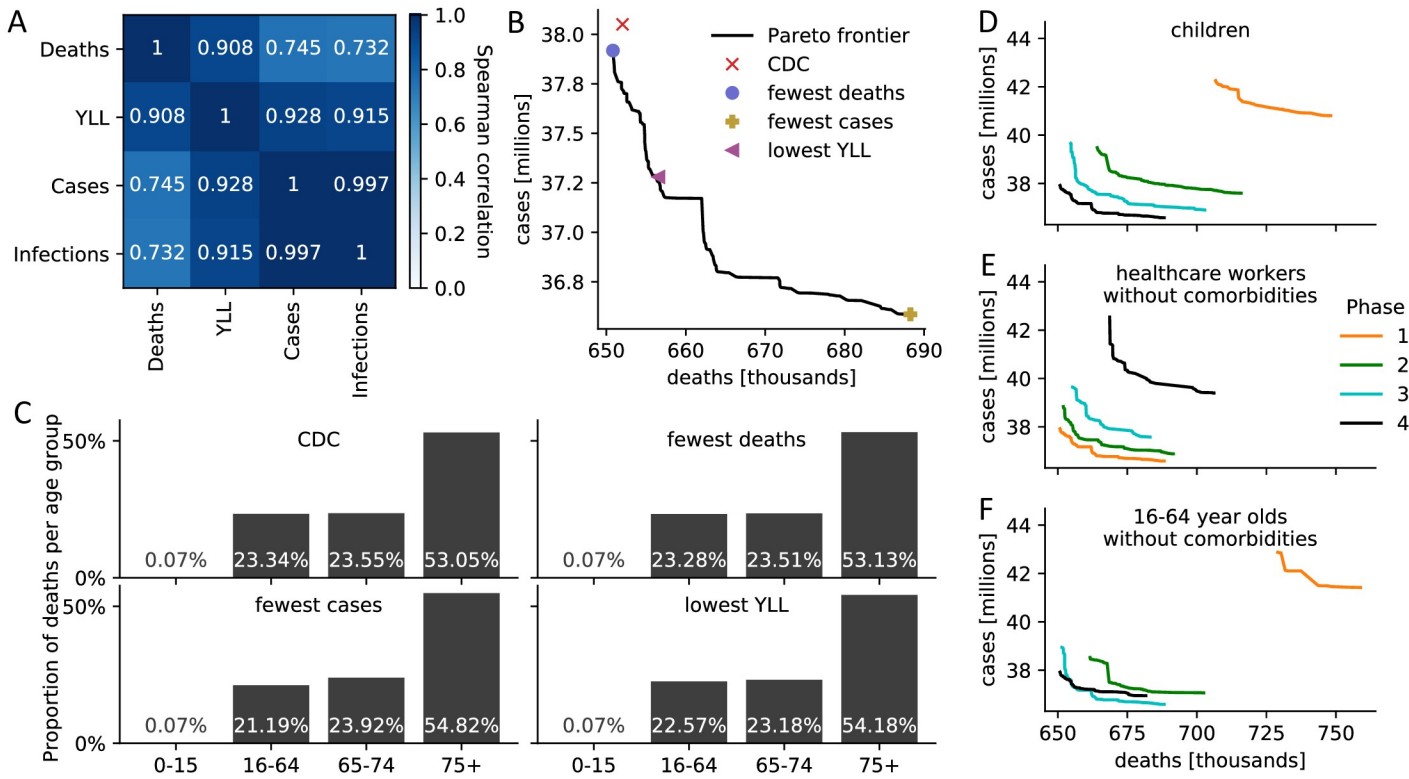

**Fig 2. Comparison of CDC and optimal vaccine allocation strategies.** (A) Spearman correlation between four measures of disease burden based on a complete comparison of all 17.5 million meaningful four-phase vaccine allocation strategies. (B) Pareto frontier of all optimal strategies based on a global search of all 17.5 million meaningful vaccine allocation strategies. For strategies on the Pareto frontier, there exists no other strategy that performs better in one objective (minimizing deaths or cases) while not performing worse in the other objective. The death and case count resulting from four specific allocations is highlighted. (C) For the four strategies highlighted in (B), the distribution of all resulting deaths across the four age groups is shown as a measure of equity. (D-F) Pareto frontiers of all optimal strategies are shown when restricting (D) children, (E) healthcare workers without comorbidities, (F) 16–64 year old without comorbidities and without an essential occupation to a certain priority phase. S8 Fig contains Pareto frontiers for all sub-populations.

Vaccinating children in any but the last phase always led to a worse outcome, irrespective of the specific objective (Table 2 and Fig 2D). The CDC prioritization of healthcare workers is validated by the model (Fig 2E). Similarly, the CDC assignment of the general public, 16–64 year-olds with non-essential occupations and no comorbidities, to the last phase is also validated (Fig 2F), with the following exception: If solely incidence (i.e., cases) is minimized irrespective of YLL and mortality, then this group (sub-population 8 in Table 2) should be vaccinated before older individuals with comparably fewer contacts (sub-populations 12–17). Interestingly, the CDC allocation is very similar to the allocation that minimizes mortality except for the phase assignment of three older age groups with no comorbidities, two of which are in congested living; these sub-population are vaccinated earlier under the optimal mortality allocation (Table 2).

## Sensitivity of results to unknown parameters

Several parameters related to virus spread and vaccine function are still unknown [27]. We therefore investigated the robustness of our findings when the relative contagiousness of asymptomatic (compared to symptomatic or pre-symptomatic), $f_A$, the relative contagiousness of vaccinated (compared to non-vaccinated) individuals who are infected, $f_V$, and the age-dependent clinical fraction (scaled by $q_{75+}$) were varied. Overall, variation of a single parameter

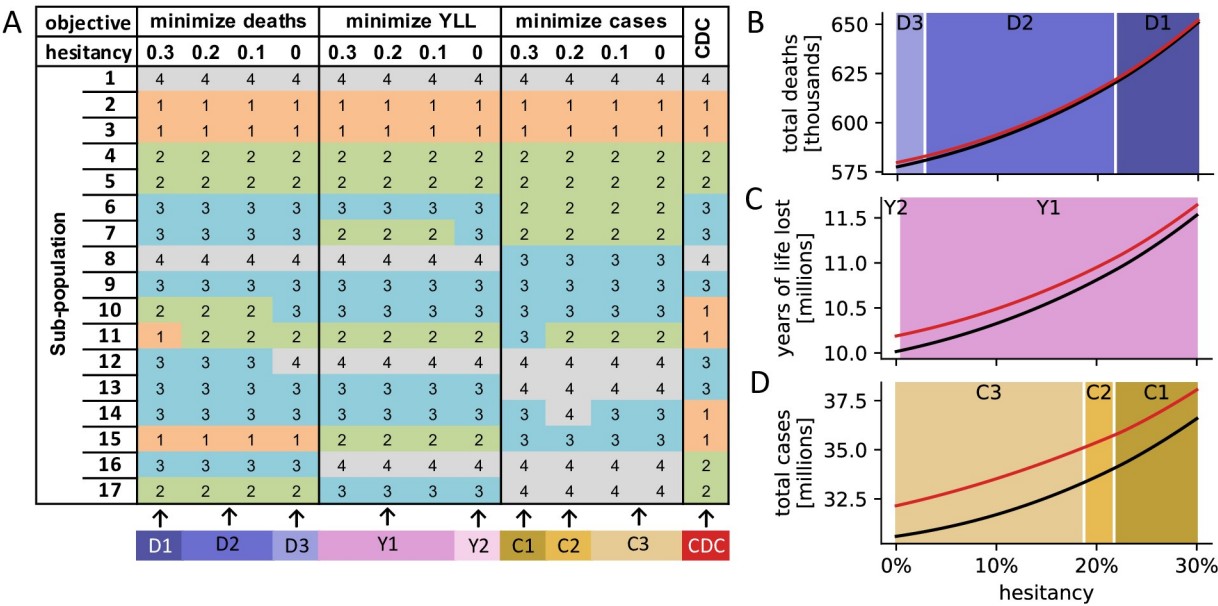

**Fig 3. Dependency of findings on vaccine hesitancy.** (A) For different levels of vaccine hesitancy (30%, 20%, 10%, 0%), the optimal vaccine allocation strategies with respect to three objectives (top row) are shown, in addition to the CDC allocation. Sub-populations 1–17 are defined as in Table 2. (B-D) Comparison of outcomes (total deaths (B), years of life lost (C) and total cases (D)) when using the respective optimal allocation strategy from (A; black line) and the CDC strategy (red line), for any vaccine hesitancy between 0% and 30%. The background color indicates which of the allocation strategies, identified in (A; bottom row), was optimal for a specific level of hesitancy.

only led to slight changes in the optimal allocations (S3 Table). Interestingly however, if vaccination does not reduce virus spread ($f_V = 1$) other than through a reduction of infections, prioritization of elderly people becomes less important as the vaccine does not act as a barrier to infection among the elderly. This is evident from S3 Table when comparing the phase assignment of individuals 65 and older in the scenarios $f_V = 0$ and $f_V = 1$: In the latter scenario, several elderly sub-populations get vaccinated later when minimizing either deaths or cases.

We next investigated if variation in the population-wide level of vaccine hesitancy affects allocation priorities. We computed the optimal vaccine allocation under four different levels of vaccine hesitancy (30%, 20%, 10%, 0%). Despite minor variations, the optimal strategies for each objective were mostly consistent across all levels of vaccine hesitancy (Fig 3A). Similarly, for any level of vaccine hesitancy between 0% and 30%, the relative differences in outcomes between the CDC and the respective optimal allocation were also comparable (Fig 3B–3D). As expected, both the CDC and the respective optimal allocation led to a worse outcome the higher the level of hesitancy in the population. In the absence of vaccine hesitancy, the United States using the CDC allocation strategy would have suffered 579,804 COVID-19-related deaths at the end of 2021, compared to 652,043 when 30% of the population refuse the vaccine.

## Impact of vaccine function

While clinical trials provide a good estimate of vaccine effectiveness (around 90% for both the Pfizer-BioNTech and Moderna vaccine, the first two vaccines used in the United States [34]), vaccine function is less well understood. We therefore examined the dependency of our findings on the way a vaccine works: through a reduction of infections ($\sigma$), and/or a reduction of symptomatic infections and a proportional increase of truly asymptomatic infections ($\delta$). A longitudinal UK COVID-19 infection study indicated $\sigma = 70\%$ and $\delta = 67\%$ for the

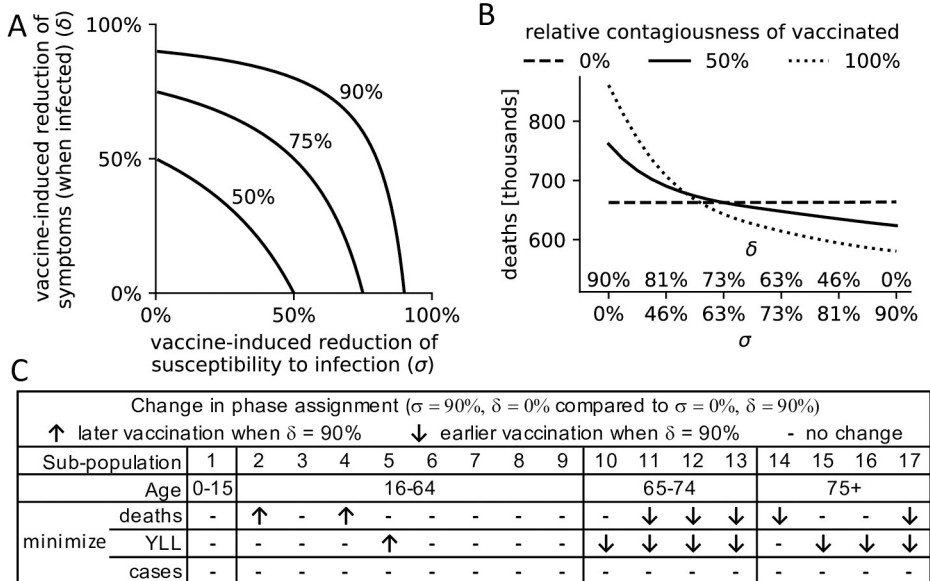

**Fig 4. Dependency of findings on vaccine function.** (A) A continuum of combinations of $\sigma$, the reduction of infections among vaccinated, and $\delta$, the reduction of symptomatic infections among vaccinated infected individuals, can lead to a vaccine effectiveness of 50%, 75% or 90%. (B) Total deaths (y-axis) under a variety of scenarios, assuming a vaccine effectiveness of 90% and the use of the CDC allocation strategy. Scenarios differ in the relative contribution of $\sigma$ and $\delta$ to the vaccine effectiveness (x-axis, see (A)), and the relative contagiousness of vaccinated individuals (compared to non-vaccinated), specified by line type (dashed: 0%, solid: 50%, dotted: 100%). (C) For three different objectives, the optimal vaccine allocation strategies are compared between two vaccines of extreme function: a vaccine that solely prevents infections ($\sigma = 90\%$, $\delta = 0\%$) and a vaccine that solely prevents symptoms among infected individuals (($\sigma = 0\%$, $\delta = 90\%$). Sub-populations (defined as in Table 2) that are allocated to a later (earlier) priority phase in the latter vaccine are indicated by $\uparrow$ ($\downarrow$). S4 Table contains the specific phase assignments for each sub-population.

AstraZeneca and Pfizer-BioNTech vaccines (the values used throughout this study) [35]. However, vaccine function can differ strongly between vaccines, as a continuum of combinations of these two parameters can give rise to the same vaccine effectiveness (Fig 4A).

In addition, it is currently unknown how contagious vaccinated infected individuals are compared to non-vaccinated individuals. Deaths under the CDC allocation were highest in scenarios where vaccinated individuals were relatively more contagious and when a vaccine was less effective at preventing infections and more effective at reducing symptoms (Fig 4B). In scenarios with a higher relative contagiousness of vaccinated individuals, vaccine function had a stronger influence on mortality. However, if vaccinated individuals were assumed to not cause any new infections, mortality was similar no matter if the vaccine solely prevented infections ($\sigma = 90\%$, $\delta = 0\%$) or solely prevented symptomatic infections among vaccinated infected individuals ($\sigma = 0\%$, $\delta = 90\%$).

Vaccine function did not affect the choice of optimal allocation when minimizing incidence (Fig 4C and S4 Table). When minimizing mortality, however, vaccine function mattered. The less effective a vaccine was at preventing infections (and correspondingly the more effective at reducing symptoms) the more vaccination prioritizations shifted towards the elderly and comorbid population, to the extent that e.g. healthcare workers without comorbidities were no longer part of the first phase in the extreme case of a vaccine that only prevents symptoms ($\sigma = 0\%$). The general trend towards the elderly and comorbid population was also prevalent when minimizing YLL.

## Discussion

Any vaccine allocation strategy must balance several competing goals, including minimizing mortality and infections, ensuring equity across demographic groups, and maintaining health care capacity. Overall, the CDC allocation performed well in all metrics utilized in this study (fewest deaths, lowest YLL, fewest infections, fewest cases); for each metric, the CDC allocation was within 5% of the respective single-objective optimal allocation. Single-objective optimal allocations tended to do poorly in other objectives, with the optimal YLL strategy best balancing trade-offs associated with minimizing mortality versus incidence (Table 2).

The most equitable allocation across age groups in terms of mortality performed poorly in all other objectives (S7 Fig). The allocation that minimized overall mortality led to a more even distribution of deaths across age groups than any other single-objective optimal allocation, and led to almost identical equitability as the CDC allocation (e.g., 53.05% vs. 53.13% of deaths occurred within the 75+ age group; Fig 2C). Our model also validates the CDC allocation with respect to maintaining health care capacity [36]. All optimal strategies agree with the CDC allocation in ranking healthcare workers as a higher priority than frontline essential workers, who are a higher priority than other essential workers. In general, the optimal allocation strategy depends strongly on the goal of the vaccination campaign: for example, sub-population 15 (age 75+ in congested living with comorbidities) should be vaccinated in the first phase to minimize mortality, in the second phase to minimize YLL, in the third phase to minimize cases, and in the last phase to minimize infections (Table 2).

The CDC allocation did not include children under 16 as part of its phased allocation scheme, as clinical trials leading to the first Emergency Use Authorization authority (EUAs) did not involve children [2], and most children seem at low risk for complications from COVID-19 [37]. However, this demographic comprises approximately 20% of the U.S. population and through social contact with adults can act as disease vectors. Our model therefore investigated the potential for possible indirect and/or cascade effects across the entire population if children under 16 were not targeted for priority vaccination. Our model validated the CDC's decision—vaccinating children in any but the last phase always led to a worse outcome (Fig 2D).

Our model included several dynamic elements of the COVID-19 pandemic that were important for a good fit to the data. First, the overall social distancing level at a particular time depended on the number of current active cases. We modeled this response through a Hill function, which allows for an initially slow response to changing case numbers reported by the media, followed by a strong response that eventually tapers off (S3 Fig). This approach does not account for changes in social distancing behavior over time, as, for example, people experiencing quarantine fatigue may increase their social interactions later in the pandemic even when case counts are high.

The model included two age-dependent biological parameters: the susceptibility to infection and the fraction of cases that develop symptoms (clinical fraction; S1 Fig). We modeled both parameters as linearly increasing with age, and fitted the former parameter to observed cumulative cases and deaths, using a weighted least squares approach that places larger weights on more recent data. This choice of weights ensures that the fit is good toward the end of the time series in order to obtain more realistic model predictions. Note that fitting both parameters at the same time is not possible given only death and case counts (over-fitting), which is why we conducted uncertainty analyses to examine the effect of variation of the age-dependent clinical fraction on the choice of optimal allocation strategies 1.

Finally, a time-dependent transmission rate accounted for the emergence of more transmissible variants of SARS-COV-2. The relative transmission rate was calculated based on biweekly estimates of the prevalence of variants in the population and the estimated relative

transmission rate of the circulating variants (S5 Fig). These four dynamic model elements—case-dependent social distancing level, age-dependent susceptibility to infection, age-dependent clinical fraction, and time-dependent transmission rate—were essential to capturing realistic dynamics in our model (S6 Fig). By identifying these elements as essential to the modeling process, we have also identified important aspects of COVID-19 epidemic dynamics that warrant further study, and should be included in realistic models of COVID-19 epidemiology.

Our model suggests one improvement that could be made to future vaccination allocations should they become necessary. The allocations identified as optimal for minimizing deaths and YLL distinguished between people with and without comorbidities in all age groups, and assigned priority to those with comorbidities. The CDC allocation only prioritizes people with comorbidities among the general working age population, and while accounting for comorbidity status in older populations leads to better model outcomes, segregating these populations may be impractical, especially in congested living conditions.

The level of vaccine hesitancy affected optimal vaccine allocation strategies only slightly (Fig 3). The CDC allocation was most similar to the allocation that minimized mortality irrespective of the level of hesitancy, and generally as hesitancy decreased the difference in outcomes between the CDC allocation and the respective optimal allocation increased. That is, the CDC allocation performs closest to optimality at the high U.S. estimate of 30% vaccine hesitancy [32].

One limitation of this study is the simplifying assumption that all sub-populations exhibit the same level of vaccine hesitancy, and that hesitancy does not change over time. Incorporating these additional dynamic elements into the model would improve the accuracy of the results, but would significantly increase model complexity.

Further limitations stem from uncertainties regarding key model parameters. Rates of contact between individuals of different age groups were based on extensive pre-pandemic survey work in eight European countries and inferred for the United States [20, 21]. In the absence of data, we assumed that contacts within a group in a congested living situation occur at double the rate of the same age group not living in congested conditions, and the additional contacts are with other individuals in the same congested living situation. Model results depend strongly on the contact matrix, and better information about contact rates, especially during the pandemic, could improve the accuracy of model predictions. The contagiousness of asymptomatic as well as vaccinated individuals is still not well understood. The model results were however robust to uncertainty in these parameters (S3 Table): variation in both phase assignments and the overall shape of the Pareto frontier were small.

We did not consider reinfections in our model. Repeated infections have been reported in the literature [38], but they seem rare and a recent study suggests prolonged immunity in most successfully vaccinated or previously infected individuals [39]. We further assumed that individuals were immediately fully protected once they received their first vaccine dose. While this model simplification overestimates the immediate effect of the vaccine, it does so uniformly and should thus not affect relative comparisons between allocation strategies.

In conclusion, the CDC allocation strategy performed well in all considered vaccination goals but never optimally, and the CDC allocation was most similar to the optimal allocation strategy that minimizes mortality. Our model validates the CDC allocation strategy with respect to equity across age groups, maintaining health care capacity, and assigning children under the age of 16 to the lowest-priority vaccination phase. Vaccine strategies that prioritize individuals with comorbidities led to slightly better outcomes than the CDC allocation strategy. The developed global optimization approach can be used to inform the design of future vaccine allocation strategies in the United States and elsewhere.

## Detailed methods

### Compartmental disease model

We used a compartmental disease model comprised of a system of deterministic ordinary differential equations to depict the dynamics of the COVID-19 epidemic in the United States. Every individual is either susceptible to the virus (S), recently infected but not yet spreading the virus (E: exposed), not yet showing symptoms but spreading the virus (P: pre-clinical), showing symptoms and spreading the virus (C: clinical), showing symptoms but not spreading the virus due to isolation or hospitalization (Q: quarantine), infected but asymptomatic (i.e., never showing symptoms) and spreading the virus (A), recovered (i.e., no longer spreading virus) after having had symptoms (RC), recovered after an asymptomatic course of infection (RA) or dead (D). In addition, every individual is either vaccinated (V), willing to be vaccinated (W) or not willing to be vaccinated (N). Combined, this leads to 20 different compartments:

$$S^N, S^W, S^V, E^N, E^W, E^V, A^N, A^W, A^V, RA^N, RA^W, RA^V, P^N, P^W, P^V, C, C^V, Q, RC, D.$$

We assume that individuals who exhibit COVID-19 symptoms are not being vaccinated, therefore we only distinguish between vaccinated and non-vaccinated individuals in the clinical compartment. Due to the initial shortage of vaccines we also assume that people who have recovered from a symptomatic COVID-19 infection do not get vaccinated, since recovered individuals have some immunity against the disease. People currently in quarantine and people who have died from the disease also do not get vaccinated, therefore we only used one compartment for each of these three groups (Q, RC, D).

The model parameters governing transitions between compartments depend on an individual's characteristics (Fig 1). We therefore divided the population into different classes based on age (four classes: 0–15, 16–64, 65–74, 75+ years old), co-morbidity status (two classes: none or at least one known risk factor associated with more severe COVID-19 disease other than age), job type of the working-age population (four classes: healthcare workers, frontline essential workers, other essential workers, and all remaining people), and living situation for the elderly population of ages 65 and older (two classes: congested living or not). As of May 10, 2021, children under the age of 16 were not eligible for vaccination, and are not stratified by co-morbidity status. This leads to a total of 17 sub-populations (see Table 2) and $17 \times 20 = 340$ different compartments, each governed by a differential equation (see Model equations).

Based on 2019 data from the U.S. Census Bureau, we used a total population size of $N = 328, 239, 523$ [25]. 64.47 million are children under the age of 16, 209.46 million are 16–64 years old (an estimated 21 million health care personnel, 30 million frontline essential workers, 20 million other essential workers [2]), 31.58 million are 65–74 years old and 22.5 million are 75 and older. Around 1.35 million people live in nursing homes and 65.3% of them are 75 and older [40]. Due to unavailability of data, we assumed the same age distribution for individuals living in congested long-term care facilities, yielding an estimated 1.04 million 65–74 year old and 1.96 million 75+ year old in congested living conditions. Using published population-level estimates and U.S. census data, we inferred the proportion of individuals with co-morbidities to be 36.72%, 73.15%, and 80.18% for the age groups 16–64, 65–74, and 75+, respectively [19, 25]. Altogether, this yields the number of people in each of the 17 sub-populations, denoted $N_i$ and shown in Table 2.

### Infection

The force of infection depends on the number of contagious individuals, their contagiousness, the age-dependent susceptibility to infection and the contact rates between the various sub-

populations. The relative contagiousness of asymptomatically infected individuals, denoted $f_A$, is a heavily debated element in the spread of COVID-19 [27, 41]. Following the best estimate of the CDC and the Office of the Assistant Secretary for Preparedness and Response (ASPR) we assumed that asymptomatically infected individuals are 25% less contagious than pre-symptomatic and symptomatic individuals [27]. In sensitivity analyses, we varied this number from 25% to 100%, the CDC's lower and upper bound for that reduction. Similarly, we assumed that vaccinated infected individuals may have lower contagiousness, denoted by $f_V$ where $f_V = 1$ implies equal contagiousness as non-vaccinated infected individuals (see Vaccine function for details).

Previous work indicates that the rate of susceptibility to infection may vary with age [17]. Here, we assumed that this rate depends linearly on age, while it does not depend on job type, living situation or comorbidity status (these latter three characteristics are incorporated into the model via the interaction matrix, see Contact rates). That is,

$$\beta_i = b_0 + b_1 \cdot \text{mean age of sub-population i}, \tag{1}$$

for $i = 1, \ldots, 17$, where $b_0$ and $b_1$ are parameters, which we fitted using the observed cumulative cases and deaths between December 14, 2020 and April 29, 2021 (see Model calibration for details).

To account for the emergence of variants with up to 50% increased infectivity, we included in the model a time-dependent relative infectivity $\phi(t) \in [100\%, 150\%]$, which we calculated from CDC genomic surveillance data as follows [42]. For each available time point, we computed the average infectivity based on the prevalence of variants B.1.1.7 and B.1.351 with an estimated 50% increased transmissibility [23, 43], variants B.1.427 and B.1.429 with an estimated 20% increased transmissibility [44], and the rest with standard transmissibility (S5 Fig).

The force of infection for sub-population $i$, $i = 1, \ldots, 17$ at time $t$, is then given by

$$\Lambda_i = \phi(t)(1 - r(\text{active cases}))\beta_i \cdot$$
$$\cdot \sum_{j=1}^{17} X_{ij}((f_A(A_j^N + A_j^W + f_V A_i^V) + P_j^N + P_j^W + f_V P_j^V + C_j + f_V C_j^V))/N_j,$$

where $r(\text{active cases})$ represents the overall social-distancing level based on the current number of active cases (see Population-wide social distancing level), and $X_{ij}$ denotes the average daily number of contacts an individual in sub-population $i$ has with individuals from sub-population $j$.

## Contact rates

Rates of contact between individuals of different ages have been identified through extensive survey work in eight European countries and subsequently inferred for a total of 152 countries including the United States [20, 21]. The age-to-age interactions in these sources are provided for 5-year age groups (e.g., $0 - 4, 5 - 9, 10 - 14, \ldots$ years of age). Using 2019 U.S. census data [25], we transformed the original contact matrix into a $4 \times 4$-contact matrix with age groups corresponding to those used in this study ($0 - 15, 16 - 64, 65 - 74, 75+$ years of age; S2A and S2B Fig). As they are based on empirical survey data, the original contact matrices are typically not symmetric. We therefore symmetrized the $4 \times 4$-contact matrix using an established procedure [45] (S2C Fig). Finally, we expanded the $4 \times 4$-contact matrix into a $17 \times 17$-contact matrix, which describes the rates of contact among the 17 sub-populations (S2D and S2E Fig), and adjusted this matrix for differences in contact rates due to the types of jobs and living conditions as follows (S2F and S2G Fig).

A recent multivariate statistical analysis found that healthcare workers have a 3.4-fold increase in their risk of infection compared to the general population [22]. Assuming this

increased risk is due to increased contacts, we calculated the relative differences in overall contacts for healthcare workers (sub-populations 2&3) compared to the general 16–64 year-old population (sub-populations 8&9). In the absence of data, we assumed that frontline essential workers (sub-populations 4&5) and other essential workers (sub-populations 6&7) have fewer contacts than healthcare workers but more than the general public, and we assumed a linear trend. The relative number of overall contacts for the different job types (healthcare workers, frontline essential workers, other essential workers, and the remainder) is therefore [3.4, 2.6, 1.8, 1], respectively. Using the sub-population sizes as weights [46], we then calculated the absolute daily contact numbers for the different job groups so that the overall total contact rate is not changed (S2F Fig).

In the absence of data, we assumed that people in congested living conditions (sub-populations 10&11 and 14&15) have double the number of contacts than peers of the same age group (sub-populations 12&13 and 16&17). We assumed that all increased contacts happen within the congested living environment and increased the respective entries proportional to their relative values. In the end this procedure yielded a 17 × 17 contact matrix that describes the rates of contact between the different sub-populations (S2G Fig).

## Population-wide social distancing level

To adjust for various levels and intensities of lockdowns and other non-pharmaceutical interventions aimed at reducing virus spread, we included a variable social distancing level that depends on the current number of active cases, $\sum_{i=1}^{17}(C_i + C_i^V)$. In particular, we used a Hill function to model the contact reduction,

$$r(\text{active cases}) = 1 - \frac{1}{1 + \left(\frac{c}{\log_{10}(\text{active cases})}\right)^k}, \tag{2}$$

where $c$ and $k$, the two parameters governing the shape of this function, were fitted to the observed cumulative cases and deaths between December 14, 2020 and April 29, 2021 (see Model calibration and S3 Fig).

## Model dynamics and disease parameters

Non-vaccinated susceptible individuals ($S_i^N$ and $S_i^W$) become infected at a rate of $\Lambda_i$. This rate is reduced by a factor of $\sigma$ for vaccinated individuals. Upon infection, susceptible individuals move into the respective exposed compartment ($E_i^N$, $E_i^W$ and $E_i^V$). Individuals remain in the exposed compartment for an average of $1/\mu_E = 3.7$ days [26]. After this incubation period, individuals start to spread the virus. A fraction $q_i$ of exposed individuals (called the clinical fraction) becomes pre-clinical ($P_i^N$, $P_i^W$ and $P_i^V$), while the others will never develop symptoms and remain asymptomatic ($A_i^N$, $A_i^W$ and $A_i^V$). The rate of truly asymptomatic infections is still not well known [41]. We assumed that the clinical fraction changes linearly with age:

$$q_i = q_{75+} - \gamma \cdot \text{mean difference in age between age group 75+ and sub-population i}, \tag{3}$$

for $i = 1, \ldots, 17$, where $q_{75+}$ is the probability of symptomatic infection for age group 75+ and $\gamma$ is chosen such that the overall (expected) clinical fraction across all ages is 70%, the CDC's most likely estimate [27]. In the model, we used $q_{75+} = 85\%$ as the base value and considered $q_{75+} = 70\%$ and $q_{75+} = 100\%$ in uncertainty analyses (S1 Fig).

Pre-clinical individuals start exhibiting symptoms after an average of $1/\mu_P = 2.1$ days and move into the clinical compartments, $C_i$ and $C_i^V$ [17]. Symptomatic individuals continue to spread the virus for an average of $1/\mu_C = 2.72$ days—the average time between the reported

onset of symptoms and a positive test in individual case data released by the CDC [47]. We assume that the clinical cases stop spreading the disease due to either isolation (moving into compartment $Q_i$). After an average of $1/\mu_Q = 19.28$ days, individuals either recover or die. A sub-population-dependent case fatality rate $CFR_i$ describes the fraction of individuals who die (move into compartment $D_i$; see Case fatality rates for details). We used $1/\mu_C + 1/\mu_Q = 22$ days for the average time from symptom onset to death as the best fit for the delay between the curves of reported U.S. cases and deaths [28].

Asymptomatic individuals (in $A_i^N$, $A_i^W$ and $A_i^V$) spread the virus at a lower rate than symptomatically infected. The parameter $f_A$ describes the relative contagiousness of asymptomatic individuals, compared to individuals in the pre-clinical and clinical compartments. We used $f_A$ = 75% and varied this important parameter from 25% to 100% in sensitivity analyses [27] (S3 Table). Asymptomatic individuals spread the virus for an average of $1/\mu_A = 5$ days [17], after which they move into their corresponding recovered compartment, $RA_i^N$, $RA_i^W$ or $RA_i^V$. We distinguish between recovered individuals who were symptomatically and asymptomatically infected because the latter may receive a vaccine since we did not consider seropositivity tests prior to vaccination. All individuals willing to receive the vaccine and without a history of symptomatic COVID-19 infection (i.e., those in $S_i^W$, $E_i^W$, $P_i^W$, $A_i^W$ and $RA_i^W$) get vaccinated at a rate $v_i(t)$, which depends on the vaccine allocation strategy and the number of daily available vaccines (see Vaccine function), and transition into the corresponding compartment upon vaccination (i.e, $S_i^V$, $E_i^V$, $P_i^V$, $A_i^V$, and $RA_i^V$).

## Case fatality rates

We calculated the sub-population-specific case fatality rates, $CFR_i$, by combining several sources. First, we divided CDC age-structured death counts by case counts to estimate the age group-dependent CFRs: 0.0129%, 0.4533%, 4.9781%, 16.7279% for the four age groups used in this study, $0 − 15$, $16 − 64$, $65 − 74$ and $75+$ years of age [48]. From existing U.S. population level estimates [19], we calculated the prevalence of comorbidities among individuals in each age group to be 18.60%, 36.72%, 73.15%, 80.18%. A study of U.S. health insurance claims indicated that 51.71% of all individuals diagnosed with COVID-19 had at least one comorbidity, while the percentage was 83.29% in individuals who died from the disease [30]. We therefore assumed that individuals with a comorbidity, irrespective of age, have a $(0.8329/0.5171)/(0.1671/0.4829) = 4.65$ times higher CFR. Combining these calcuations yields an age-dependent CFR of 0.1935%, 1.13551%, 4.2560% for adults without comorbidities and 0.8997%, 6.3012%, 19.7907% for adults with comorbidities in the age groups $16 − 64$, $65 − 74$ and $75+$, respectively. Note that in this study we do not distinguish between children with and without comorbidities, and instead use the overall CFR of 0.0129% for this age group.

## Vaccine function

Vaccinated individuals have a lower chance of developing a symptomatic COVID-19 infection. After clinical trials reported efficacy rates of around 95% [49, 50], initial data from the US vaccine roll-out found the Pfizer/BioNTech BNT162b2-mRNA and Moderna mRNA-1273 vaccines to be 90% effective [34]. This means that, all other things being equal (e.g., age, contact rates, comorbidity status), a vaccinated person is 90% less likely to develop a symptomatic COVID-19 infection than a non-vaccinated person. In a compartmental model, this reduction could be due to one or both of two mechanisms:

1. a vaccine-induced reduced susceptibility to infection, i.e., a reduction in the number of individuals who move from the S compartment to the E compartment. We denote this reduction factor by $\sigma \in [0, 1]$, or

2. a vaccine-induced reduced probability of developing a symptomatic infection (when infected), i.e., a reduction in the number of individuals who move from the E compartment to the P compartment, with a corresponding increase in the number of individuals who move from the E compartment to the A compartment. We denote this reduction factor by $\delta \in [0, 1]$.

Vaccine effectiveness and the two reduction factors are related by:

$$\text{vaccine effectiveness} = 1 - (1 - \sigma)(1 - \delta).$$

Fig 4A shows all possible combinations of reduction factors that yield a particular vaccine effectiveness. A longitudinal British study of the effectiveness of the AstraZeneca and the Pfizer-BioNTech vaccine suggests that $\sigma = 70\%$ and $\delta = 66.7\%$ [35]. In sensitivity analyses, we varied both values from 0% to 90% (Fig 4). In practice, these two reduction factors as well as vaccine effectiveness may differ among sub-populations (e.g., with age or comorbidity status) or among different vaccines; in this study we only investigated fixed reduction factors.

In addition, vaccinated infected individuals may spread the virus at a lower rate than non-vaccinated individuals. The reduced contagiousness of vaccinated individuals, $f_V \in [0\%, 100\%]$ accounts for this in the force of infection. $f_V = 0\%$ corresponds to a complete stop of virus spread while $f_V = 100\%$ means no reduction compared to non-vaccinated individuals. Due to a lack of data, we assumed $f_V = 50\%$ in the base model—that is, a vaccinated infected person is 50% less contagious than a non-vaccinated infected person at the same stage of the disease. In a sensitivity analysis, we varied $f_V$ from 0% to 100% (Fig 4B).

## Vaccine hesitancy

Recent data indicates that up to 30% of the population are hesitant to receive COVID-19 vaccines [31, 32]. We evaluated the CDC allocation using a base value of 30% hesitancy but also studied the effect of lower hesitancy on the choice of optimal vaccine allocations (Fig 3). That is, we set $S_i^N(0) = \text{hesitancy}(S_i^N(0) + S_i^W(0))$. Note that we assumed uniform hesitancy levels for all sub-populations.

## Vaccination campaign

The public vaccine roll-out in the United States began on December 14, 2020, denoted by $t_0$. For simplicity, we considered a single vaccination event per individual. We assumed that the number of individuals newly vaccinated at day $t$, denoted $\xi(t)$, is half the 7-day average in total doses administered, based on U.S. vaccination records released by the CDC and the two-dose vaccine regimen [33]. After May 5, 2021, we projected future daily vaccination levels as follows: Non-vaccinated individuals willing to get vaccinated will likely become increasingly harder to find. We thus assumed that the number of daily vaccinations decreases linearly at a rate such that it becomes zero exactly when all individuals willing to be vaccinated have been vaccinated (S4 Fig).

The CDC announced in December 2020 two phases of vaccine roll-out that were further divided into a total of four priority phases (1a, 1b, 1c, 2; Table 2) [2]. To enable a direct evaluation of the CDC vaccine allocation strategy, we also considered four phases, labeled 1, 2, 3, and 4, for brevity. All sub-populations $i$ that are allocated to the current priority phase receive the vaccine at the same time-dependent rate

$$v_i(t) = \frac{\xi(t)}{\sum_{\text{sub–population } k \text{ part of the current phase}} S_k^W(t) + E_k^W(t) + A_k^W(t) + RA_k^W(t) + P_k^W(t)}.$$

For all other sub-populations $j$, we have $v_j(t) = 0$. That is, all individuals without a present or past symptomatic COVID-19 infection are available to receive the vaccine, and we do not consider seropositivity tests and the corresponding exclusion of currently or previously asymptomatically infected people from vaccination. Once there are no more non-vaccinated individuals who are willing to be vaccinated and are part of a certain priority phase, the vaccination campaign moves to the next phase. Once all people willing to be vaccinated receive the vaccine the campaign stops. We do not consider reinfections, so the lack of vaccination of previously symptomatically infected individuals has no effect on the model dynamics.

## Model equations

A schematic illustration of the compartmental model is shown in Fig 1. The compartmental model is described by the following system of differential equations, where $i = 1, \ldots, 17$ enumerates the 17 sub-populations:

$$
\begin{aligned}
\dot{S}_i^N &= -\Lambda_i S_i^N \\
\dot{S}_i^W &= -\Lambda_i S_i^W - v_i S_i^W \\
\dot{S}_i^V &= -\Lambda_i (1 - \sigma) S_i^V + v_i S_i^W \\
\dot{E}_i^N &= \Lambda_i S_i^N - \mu_E E_i^N \\
\dot{E}_i^W &= \Lambda_i S_i^W - \mu_E E_i^W - v_i E_i^W \\
\dot{E}_i^V &= \Lambda_i S_i^V (1 - \sigma) - \mu_E E_i^V + v_i E_i^W \\
\dot{A}_i^N &= \mu_E (1 - q_i) E_i^N - \mu_A A_i^N \\
\dot{A}_i^W &= \mu_E (1 - q_i) E_i^W - \mu_A A_i^W - v_i A_i^W \\
\dot{A}_i^V &= \mu_E (1 - q_i(1 - \delta)) E_i^V - \mu_A A_i^V + v_i A_i^W \\
\dot{RA}_i^N &= \mu_A A_i^N \\
\dot{RA}_i^W &= \mu_A A_i^W - v_i RA_i^W \\
\dot{RA}_i^V &= \mu_A A_i^V + v_i RA_i^W \\
\dot{P}_i^N &= \mu_E q_i E_i^N - \mu_P P_i^N \\
\dot{P}_i^W &= \mu_E q_i E_i^W - \mu_P P_i^W - v_i P_i^W \\
\dot{P}_i^V &= \mu_E q_i(1 - \delta) E_i^V - \mu_P P_i^V + v_i P_i^W \\
\dot{C}_i &= \mu_P (P_i^N + P_i^W) - \mu_C C_i \\
\dot{C}_i^V &= \mu_P P_i^V - \mu_C C_i^V \\
\dot{Q}_i &= \mu_C (C_i + C_i^V) - \mu_Q Q_i \\
\dot{RC}_i &= (1 - \text{CFR}_i) \mu_Q Q_i \\
\dot{D}_i &= \text{CFR}_i \mu_Q Q_i
\end{aligned}
\tag{4}
$$

## Model calibration

Most model parameters were derived from the literature (Table 1). We used an elitist genetic algorithm [51] to estimate four model parameters by fitting the modeled to the observed cumulative cases and deaths between December 14, 2020 and April 29, 2021, obtained from the COVID-19 Data Repository at the Center for Systems Science and Engineering at Johns Hopkins University [24, 47]. Two of these parameters, $b_0 \in [0, 0.1]$ and $b_1 \in [0, 0.01]$, describe the linearly changing age-dependent rate of susceptibility to infection (Eq 1). The other two parameters, $c \in [4, 6]$ and $k \in [2, 16]$, describe the shape of the Hill function used to model the population-wide social-distancing level, which depends on the number of active cases (Eq 2).

In the genetic algorithm, we chose meaningful ranges for the parameters that ensured positive rates and probabilities in [0, 1]. We used a fitness function $f$ given by a weighted sum of squared errors (wSSE) between the observed and predicted cumulative deaths and cases,

$$f(\text{deaths, cases}) = \text{wSSE(deaths)} + \text{wSSE(cases)},$$

where

$$\text{wSSE(deaths)} = \sum_{d=\text{December 14, 2020}}^{\text{April 29, 2021}} w_d \cdot (\text{observed minus predicted deaths up to day } d)^2,$$

$$\text{wSSE(cases)} = \sum_{d=\text{December 14, 2020}}^{\text{April 29, 2021}} w_d \cdot (\text{observed minus predicted cases up to day } d)^2,$$

where we used quadratically increasing weights, $w_{\text{December 14, 2020}} = 1$, $w_{\text{December 15, 2020}} = 4$, $w_{\text{December 16, 2020}} = 9, \ldots$. This choice of weights ensures that the fit is particularly good at the end, yielding more realistic future model dynamics than, for instance, an unweighted fit to cumulative deaths would. To allow for equal weighing of the fit of deaths and case counts, we divided the observed and predicted cumulative cases by 50 (corresponding to a CFR of 2%) before calculating the wSSE of observed and predicted cumulative cases.

We let the genetic algorithm minimize the fitness function $f$ using 50 iterations and a population of 1000 parameter sets. Each iteration, we assigned the 300 best parameter sets as "parents" from which we generated 700 new "children" parameter sets using uniform crossover with a probability of 50%. In addition, we randomly mutated each parameter choice (within its respective range) with a probability of 10%, except for the top 10 parameter sets. That is, we used an elite ratio of 1%, which ensures that the best parameter sets are never lost due to random mutation.

Because genetic algorithms may get stuck at local optima, we performed 100 separate elitist genetic algorithms and used the parameter set with the overall lowest fitness function values for the study. For each scenario with different $f_A$, $f_V$ or $q_{75+}$ value, we ran 100 separate elitist genetic algorithms to obtain the respective best parameter sets. S1 Table shows the fitted parameter values under the CDC allocation strategy.

## Model implementation and outcomes

The model is implemented in Python 3.8 using the open source JIT compiler numba for improved run time. We ran the model for each of the 17.5 million possibly optimal allocation strategies. At the end of each run, we recorded the total number of (i) deaths, (ii) (symptomatic) cases and (iii) infections per age group. Based on the 2017 period life table from the U.S. Social Security Administration [52], we calculated that individuals in age groups $0-15$, $16-64$, $65-74$ and 75+ have 71.49, 41.31, 15.97 and 7.97 expected years of life left. We used

these numbers to derive, for each run, the total years of life lost (YLL) due to the COVID-19 pandemic. We also considered how deaths were distributed across the four age groups and used the Shannon entropy to summarize the variation in the distribution in a single measure of health equity in S7 Fig [53].

## Supporting information

**S1 Fig. Age-dependent probability of symptomatic infection.** The probability of symptomatic infection is shown for the different age groups (x-axis) and different scenarios (colors). The average probability of symptomatic infection is 70% in each scenario. This probability increases linearly with year of age up to a fixed value of 70% (blue x), 85% (orange circles; default), 100% (green diamonds) for the age group 75+.
(TIF)

**S2 Fig. Multi-step generation of the contact matrix.** (A) The original $4 \times 4$ U.S. age-age contact matrix inferred from survey data [20, 21] was transformed, using (B) U.S. census data, into (C) a symmetric $4 \times 4$ contact matrix [45]. Using (D) information on the number of individuals within each of the 17 sub-populations used in this study (characteristics defined in Table 2), the symmetric $4 \times 4$ contact matrix was expanded into (E) a $17 \times 17$ contact matrix. Some jobs require more physical contact than others. Inclusion of the average contact rates per job type yielded (F) an adapted contact matrix. Similarly, elderly individuals in congested living conditions have more contacts than their peers and all these increased contacts were assumed to occur within the congested living environment, which yielded (G) the final contact matrix used in this study.
(TIF)

**S3 Fig. Case-dependent contact reduction.** A variable contact reduction (Hill function) accounts for changes in the population-wide activity level based on the severity (i.e., the number of active cases) of the epidemic in the United States. The shape of the case-dependent contact reduction used in the base model (black line) is shown along with the shapes of the most extreme parameter choices allowed in the genetic algorithm (dashed lines).
(TIF)

**S4 Fig. Speed of the vaccine roll-out.** In the model, the number of newly fully vaccinated individuals each day is set to 50% of the 7-day average of the total number of administered doses (black line). Colored lines show predictions of the future speed of the vaccine roll-out for different levels of vaccine hesitancy.
(TIF)

**S5 Fig. Time-varying infectivity of the circulating virus strains.** (A) Prevalence of several variants of concern based on $> 40,000$ sequences collected through CDC's national genomic surveillance since Dec 20, 2020 and grouped in 2-week intervals [42]. (B) For the midpoint of each two-week interval, the relative infectivity of circulating virus strains based on a 50% increased infectivity for B.1.1.7 and B.1.351 and 20% increased infectivity for B.1.427 and B.1.429 is shown (orange circles). A fitted logistic equation with asymptotes at 100% and 150% projects the future relative infectivity (blue line).
(TIF)

**S6 Fig. Model fit.** The observed (dashed line) and model-predicted (solid line) cumulative deaths (A) and cases (B) are shown. The model parameters used are described in Table 1 and in the first row of S1 Table.
(TIF)

**S7 Fig. Death and case count for all vaccine allocation strategies.** The death and case count of all 17.5 million evaluated meaningful vaccine allocation strategies fall within the dotted region. For strategies on the Pareto frontier (solid line), there exists no other strategy that performs better in one objective (minimizing deaths or cases) while not performing worse in the other objective. The death and case counts resulting from six specific allocations are highlighted.
(TIF)

**S8 Fig. Pareto frontiers when fixing one sub-population's priority phase.** Each subpanel shows four Pareto frontiers. For each frontier, one sub-population's priority phase is fixed (see Table 2 for group characteristics). For strategies on the Pareto frontier, there exists no other strategy that performs better in one objective (minimizing deaths or cases) while not performing worse in the other objective.
(TIF)

**S1 Table. Parameters associated with the best fits for the different scenarios.** For each scenario (described by the parameters in the three most left columns), 100 separate elitist genetic algorithms were performed and the parameters associated with the best fit are shown, in addition to the value of the cost function (wSSE) that the algorithm minimized.
(TIF)

**S2 Table. Allocation strategies that outperform the CDC allocation in all three objectives.** This table shows the CDC allocation (first row) and all allocation strategies on the three-dimensional Pareto frontier that lead to fewer deaths, cases and YLL at the same time (bottom 28 rows). Sub-populations 1–17 are defined as in Table 2; sub-populations with comorbidities are highlighted in yellow.
(TIF)

**S3 Table. Comparison of CDC and optimal allocation strategies for seven scenarios.** For each sub-population (characteristics and population sizes defined in the left columns) and seven combinations of unknown disease parameters ($q_{75+}$, the proportion of symptomatic infections among individuals 75 and older; $f_A$, the relative contagiousness of asymptomatic infected individuals; $f_V$, the relative contagiousness of vaccinated infected individuals), the priority phase corresponding to the optimal allocation strategy is shown. At the bottom, predicted outcomes (deaths, YLL and cases) resulting from the CDC allocation and the respective optimal allocation strategy are compared.
(TIF)

**S4 Table. Variation of optimal allocation strategies with vaccine function.** For different types of vaccines with 90% effectiveness (specified by $\sigma$ vs $\delta$), the optimal vaccine allocation strategies with respect to three objectives (top row) are shown. Sub-populations 1–17 are defined as in Table 2.
(TIF)

## Author Contributions

**Conceptualization:** Md Rafiul Islam, Tamer Oraby, Claus Kadelka.

**Formal analysis:** Md Rafiul Islam, Claus Kadelka.

**Investigation:** Md Rafiul Islam, Tamer Oraby, Mohammad Mihrab Chowdhury, Claus Kadelka.

**Methodology:** Md Rafiul Islam, Claus Kadelka.

**Software:** Claus Kadelka.

**Visualization:** Claus Kadelka.

**Writing – original draft:** Md Rafiul Islam, Tamer Oraby, Audrey McCombs, Michael G. Tyshenko, Claus Kadelka.

**Writing – review & editing:** Md Rafiul Islam, Tamer Oraby, Audrey McCombs, Mohammad Mihrab Chowdhury, Mohammad Al-Mamun, Michael G. Tyshenko, Claus Kadelka.

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
