## [Decision Letter · Decision Letter 0]

13 Oct 2021

PONE-D-21-26548Evaluation of the United States COVID-19 Vaccine Allocation StrategyPLOS ONE

Dear Dr. Kadelka,

Thank you for submitting your manuscript to PLOS ONE. After careful consideration, we feel that it has merit but does not fully meet PLOS ONE’s publication criteria as it currently stands. Therefore, we invite you to submit a revised version of the manuscript that addresses the points raised during the review process.

We look forward to receiving your revised manuscript.

Kind regards,

Martial L Ndeffo Mbah, Ph.D

Academic Editor

PLOS ONE

Journal Requirements:

Additional Editor Comments:

Please, address the handful of minor comments and suggestions of Reviewer 2.

Reviewers' comments:

Reviewer's Responses to Questions

**Comments to the Author**

1. Is the manuscript technically sound, and do the data support the conclusions?

Reviewer #1: Yes

Reviewer #2: Yes

2. Has the statistical analysis been performed appropriately and rigorously? 

Reviewer #1: Yes

Reviewer #2: Yes

3. Have the authors made all data underlying the findings in their manuscript fully available?

Reviewer #1: No

Reviewer #2: No

4. Is the manuscript presented in an intelligible fashion and written in standard English?

Reviewer #1: Yes

Reviewer #2: Yes

5. Review Comments to the Author

Reviewer #1: In the present paper, the authors present their study about evaluating the Vaccine allocation strategy implemented by the Centers for Disease Control (CDC) in the US. This subject is of great importance in present times since the vaccine administration has started worldwide. Therefore, it is crucial to have instruments to monitor the vaccination campaign.

The study design and methodology are well structured and clearly explained. The results and conclusions are well presented and thoroughly discussed. Therefore, the present study may be of great help in optimising the vaccination strategy in different countries.

The description of the compartmental model and the calibration procedure is very precise, and the discussion of the parameters is exhaustive. Overall, the paper is well written and well organised.

My only concern is relative to the code availability; no Python code is available at

the address https://github.com/ckadelka/COVID19-CDC-allocation-evaluation reported in the text.

Therefore, the decision is to accept the paper in the present form only after the python source codes are available.

Reviewer #2: Using mathematical modeling, this manuscript analyzes if the strategy of vaccine allocation followed in the United States was suitable in terms of certain objective functions such as minimizing the number of deaths and the number of cases. The authors find that such strategy was not precisely optimum but performed similarly to the optimum strategies (for each objective function).

In general, the manuscript is very well written, complete, and relevant. I am convinced that the translation of this model to other scenarios (for example other epidemic episodes ad regions) can be extremely helpful to guide public health policy in vaccination matters.

As any modelling exercise, the authors are forced to propose simplifications and state uncorroborated assumptions to arrive to solutions. This is perfectly understandable. To me, the assumptions of the model seem appropriate.

In sum, I find this paper truly interesting and well executed. Congratulations.

Some recommendations follow:

(a) To me, the clinical and epidemiological information presented in table 1 (i.e., info relevant to set the values of the model parameters) should be presented within the main text.

(b) Similarly, I recommend moving the scheme in S1 Figure to the main text.

(c) Line 105: “If solely incidence is minimized irrespective of YLL and mortality, then this group 105 should be vaccinated before older individuals with comparably fewer contacts”. Please clarify/elaborate.

(d) Line 118: “Interestingly however, if vaccination does not reduce virus spread (fV = 1) other than through a reduction of infections, prioritization of elderly people becomes 119 less important as the vaccine does not act as a barrier to infection among the elderly”…. This statement unclear to me. Please elaborate and discuss….

(e) Line 146: The following statement can be misleading: “The higher the relative contagiousness of vaccinated individuals, the higher was the influence of vaccine function on mortality; if vaccinated individuals did not cause any new infections, vaccine function had no noticeable effect on mortality”. This can be understood in a general sense: vaccination does not decrease mortality. Please reformulate to convey the sense of comparison….

6. PLOS authors have the option to publish the peer review history of their article (what does this mean?). If published, this will include your full peer review and any attached files.

Reviewer #1: No

Reviewer #2: No

---

## [Author Response · Author response to Decision Letter 0]

14 Oct 2021

Please see the attached Word document, which distinguishes reviewer comments and our response and edits by color.

Here, the plain txt version:

Reviewer #1: In the present paper, the authors present their study about evaluating the Vaccine allocation strategy implemented by the Centers for Disease Control (CDC) in the US. This subject is of great importance in present times since the vaccine administration has started worldwide. Therefore, it is crucial to have instruments to monitor the vaccination campaign.

The study design and methodology are well structured and clearly explained. The results and conclusions are well presented and thoroughly discussed. Therefore, the present study may be of great help in optimising the vaccination strategy in different countries.

The description of the compartmental model and the calibration procedure is very precise, and the discussion of the parameters is exhaustive. Overall, the paper is well written and well organised.

My only concern is relative to the code availability; no Python code is available at

the address https://github.com/ckadelka/COVID19-CDC-allocation-evaluation reported in the text.

Therefore, the decision is to accept the paper in the present form only after the python source codes are available.

We thank the reviewer for their positive assessment of our work. At the same time, we apologize for the oversight on our end. We simply forgot to upload the Python code to the GitHub repository, https://github.com/ckadelka/COVID19-CDC-allocation-evaluation. It is now uploaded.

Reviewer #2: Using mathematical modeling, this manuscript analyzes if the strategy of vaccine allocation followed in the United States was suitable in terms of certain objective functions such as minimizing the number of deaths and the number of cases. The authors find that such strategy was not precisely optimum but performed similarly to the optimum strategies (for each objective function).

In general, the manuscript is very well written, complete, and relevant. I am convinced that the translation of this model to other scenarios (for example other epidemic episodes ad regions) can be extremely helpful to guide public health policy in vaccination matters.

As any modelling exercise, the authors are forced to propose simplifications and state uncorroborated assumptions to arrive to solutions. This is perfectly understandable. To me, the assumptions of the model seem appropriate.

In sum, I find this paper truly interesting and well executed. Congratulations.

We also thank this reviewer for their positive assessment of our study. Below, we describe how we addressed the particular recommendations.

Some recommendations follow:

(a) To me, the clinical and epidemiological information presented in table 1 (i.e., info relevant to set the values of the model parameters) should be presented within the main text.

We followed the reviewer’s recommendation and moved the table with the model parameters into the main text (what used to be S1 Table is now Table 1).

(b) Similarly, I recommend moving the scheme in S1 Figure to the main text.

Again, we followed the reviewer’s recommendation and moved S1 Figure to the main text (what used to be S1 Figure is now Figure 1).

(c) Line 105: “If solely incidence is minimized irrespective of YLL and mortality, then this group 105 should be vaccinated before older individuals with comparably fewer contacts”. Please clarify/elaborate.

We clarified the sentence by (i) explicitly stating that minimizing incidence is the same as minimizing cases, and (ii) explicitly referencing the specific sub-populations in Table 1 (now Table 2) that are compared here: sub-population 8, which falls into phase 3 when minimizing cases/incidence and sub-populations 12-17, which fall into phase 4. The adapted sentence now reads:

If solely incidence (i.e., cases) is minimized irrespective of YLL and mortality, then this group (sub-population 8 in Table 2) should be vaccinated before older individuals with comparably fewer contacts (sub-populations 12-17).

(d) Line 118: “Interestingly however, if vaccination does not reduce virus spread (fV = 1) other than through a reduction of infections, prioritization of elderly people becomes 119 less important as the vaccine does not act as a barrier to infection among the elderly”…. This statement unclear to me. Please elaborate and discuss….

We thank the reviewer for pointing this out. We have clarified this finding with additional explanatory sentence as follows:

Interestingly however, if vaccination does not reduce virus spread (fV = 1) other than through a reduction of infections, prioritization of elderly people becomes less important as the vaccine does not act as a barrier to infection among the elderly. This is evident from S3 Table when comparing the phase assignment of individuals 65 and older in the scenarios f_V=0 and f_V=1: In the latter scenario, several elderly sub-populations get vaccinated later when minimizing either deaths or cases.

(e) Line 146: The following statement can be misleading: “The higher the relative contagiousness of vaccinated individuals, the higher was the influence of vaccine function on mortality; if vaccinated individuals did not cause any new infections, vaccine function had no noticeable effect on mortality”. This can be understood in a general sense: vaccination does not decrease mortality. Please reformulate to convey the sense of comparison….

We agree with the reviewer: the statement may be misleading, especially when read by a non-expert who may not understand the comparison conducted here. We rewrote the statement (and also slightly adapted the previous sentence) as follows:

Deaths under the CDC allocation were highest in scenarios where vaccinated individuals were relatively more contagious and when a vaccine was less effective at preventing infections and more effective at reducing symptoms (Fig. 4B). In scenarios with a higher relative contagiousness of vaccinated individuals, vaccine function had a stronger influence on mortality. However, if vaccinated individuals were assumed to not cause any new infections, mortality was similar no matter if the vaccine solely prevented infections ($\\sigma=90\\%, \\delta=0\\%$) or solely prevented symptomatic infections among vaccinated infected individuals ($\\sigma=0\\%, \\delta=90\\%$).

---

## [Decision Letter · Decision Letter 1]

25 Oct 2021

Evaluation of the United States COVID-19 Vaccine Allocation Strategy

PONE-D-21-26548R1

Dear Dr. Kadelka,

We’re pleased to inform you that your manuscript has been judged scientifically suitable for publication and will be formally accepted for publication once it meets all outstanding technical requirements.

Kind regards,

Martial L Ndeffo Mbah, Ph.D

Academic Editor

PLOS ONE

Additional Editor Comments (optional):

Reviewers' comments:

Reviewer's Responses to Questions

**Comments to the Author**

1. If the authors have adequately addressed your comments raised in a previous round of review and you feel that this manuscript is now acceptable for publication, you may indicate that here to bypass the “Comments to the Author” section, enter your conflict of interest statement in the “Confidential to Editor” section, and submit your "Accept" recommendation.

Reviewer #1: All comments have been addressed

2. Is the manuscript technically sound, and do the data support the conclusions?

Reviewer #1: Yes

3. Has the statistical analysis been performed appropriately and rigorously? 

Reviewer #1: Yes

4. Have the authors made all data underlying the findings in their manuscript fully available?

Reviewer #1: Yes

5. Is the manuscript presented in an intelligible fashion and written in standard English?

Reviewer #1: Yes

6. Review Comments to the Author

Reviewer #1: The authors have properly addressed all the requests, therefore I suggest to publish the manuscript in the present form.

7. PLOS authors have the option to publish the peer review history of their article (what does this mean?). If published, this will include your full peer review and any attached files.

Reviewer #1: No

---

## [Editor Report · Acceptance letter]

27 Oct 2021

PONE-D-21-26548R1

Evaluation of the United States COVID-19 Vaccine Allocation Strategy

Dear Dr. Kadelka:

I'm pleased to inform you that your manuscript has been deemed suitable for publication in PLOS ONE. Congratulations! Your manuscript is now with our production department.

Kind regards,

on behalf of

Dr. Martial L Ndeffo Mbah

Academic Editor

PLOS ONE